# Destruction of Lysozyme Amyloid Fibrils Induced by Magnetoferritin and Reconstructed Ferritin

**DOI:** 10.3390/ijms232213926

**Published:** 2022-11-11

**Authors:** Jan Gombos, Lucia Balejcikova, Peter Kopcansky, Marianna Batkova, Katarina Siposova, Jozef Kovac, Kristina Zolochevska, Ivo Safarik, Alica Lokajova, Vasil M. Garamus, Dusan Dobrota, Oliver Strbak

**Affiliations:** 1Department of Medical Biochemistry, Jessenius Faculty of Medicine, Comenius University, 036 01 Martin, Slovakia; 2Institute of Hydrology, Slovak Academy of Sciences, 841 01 Bratislava, Slovakia; 3Institute of Experimental Physics, Slovak Academy of Sciences, 040 01 Kosice, Slovakia; 4Department of Nanobiotechnology, Biology Centre, ISBB, Czech Academy of Sciences, 370 05 Ceske Budejovice, Czech Republic; 5Regional Centre of Advanced Technologies and Materials, Czech Advanced Technology and Research Institute, Palacky University, 779 00 Olomouc, Czech Republic; 6Helmholtz-Zentrum Hereon, Max-Planck-Str. 1, 21502 Geesthacht, Germany; 7Biomedical Center Martin, Jessenius Faculty of Medicine, Comenius University, 036 01 Martin, Slovakia

**Keywords:** ferritin, magnetoferritin, reconstructed ferritin, lysozyme amyloid fibrils, destruction, iron release

## Abstract

Neurodegenerative disorders, including Alzheimer’s disease (AD), Parkinson’s disease (PD), or systemic amyloidosis, are characterized by the specific protein transformation from the native state to stable insoluble deposits, e.g., amyloid plaques. The design of potential therapeutic agents and drugs focuses on the destabilization of the bonds in their beta-rich structures. Surprisingly, ferritin derivatives have recently been proposed to destabilize fibril structures. Using atomic force microscopy (AFM) and fluorescence spectrophotometry, we confirmed the destructive effect of reconstructed ferritin (RF) and magnetoferritin (MF) on lysosome amyloid fibrils (LAF). The presence of iron was shown to be the main factor responsible for the destruction of LAF. Moreover, we found that the interaction of RF and MF with LAF caused a significant increase in the release of potentially harmful ferrous ions. Zeta potential and UV spectroscopic measurements of LAF and ferritin derivative mixtures revealed a considerable difference in RF compared to MF. Our results contribute to a better understanding of the mechanism of fibril destabilization by ferritin-like proteins. From this point of view, ferritin derivatives seem to have a dual effect: therapeutic (fibril destruction) and adverse (oxidative stress initiated by increased Fe^2+^ release). Thus, ferritins may play a significant role in various future biomedical applications.

## 1. Introduction

Iron is one of the most common elements on Earth and occurs in almost all forms of life as a part of many metalloproteins. It plays a unique role in many cellular processes, including cell respiration, electron transfer reactions, DNA synthesis, energy metabolism, etc. Under physiological conditions, iron usually exists in two oxidation states. The first is a relatively soluble but highly toxic ferrous form (Fe^2+^), and the second is a very insoluble but nontoxic ferric state (Fe^3+^) [1]. Ferrous ions can enter the 12 nm-sized apoferritin cage of ferritin. After oxidation, ferric ions form an antiferromagnetic ferrihydrite-like mineral core [2]. Surprisingly, the ferritin mineral core of patients with AD consists mainly of the magnetite mineral phase [3]. Unlike antiferromagnetic ferrihydrite, nanosized magnetite particles exhibit superparamagnetic properties [4]. The precursor of magnetite biomineralization is believed to be ferritin [3]. The origin of this pathological biogenic magnetoferritin formation is not fully understood.

The difference in relaxivity between native ferritin and pathological ferritin in standardized in vitro samples expands the possibilities of its application in non-invasive MRI diagnostics of neurodegenerative disorders [5]. Ferritin behavior in inflammation and other metabolic diseases is still not well understood. However, in some cases allowing iron release [6] or stored-iron reduction can eventually lead to magnetite formation. According to in vitro experiments, reducing agents (vitamins B_2_ and C) can contribute to the release of toxic ferrous ions [7]. Non-controlled iron accumulation can lead to the formation of unspecified iron compounds, which can be detected using Mössbauer spectroscopy [8]. Furthermore, differences in the quality or quantity of iron-containing biomolecules can be observed in cells, especially brain tissue [9,10]. The accumulation and biomineralization of iron oxide nanoparticles are typical signs of neurodegenerative disorders and aging [11,12]. However, magnetic nanoparticles observed in cancer tissue [13] or other disorders associated with iron accumulation can complicate accurate disease distinction. Therefore, the study of specific aggregation-prone proteins and the monitoring of iron “levels” can contribute to establishing the correct diagnosis. The formation of the *β*-sheet-rich structures, including soluble oligomers and stable/insoluble fibrillar aggregates known as *β*-plaques, may significantly affect normal neuronal functions [14,15,16,17]. From this point of view, a better understanding of fibril formation and destabilization mechanisms may help develop effective therapeutic approaches [18].

One of the well-described amyloid-prone proteins is the lysozyme. The amyloid fibrils form under destabilizing conditions, acidic pH, temperature, and agitation, and may be affected by the presence of various compounds/additives. Within the present work, we focused on the study of interactions between lysozyme amyloid fibrils (LAF) and ferritin derivatives, specifically, reconstructed ferritin (RF) represented a synthetic model system of physiological ferritin, and magnetoferritin (MF) simulated pathological ferritin. Although we previously demonstrated reductions in both magnetoferritin and reconstructed ferritin, the number of amyloid fibrils and their size in interactions with ferritin derivatives remained unrevealed [19]. Thus, to better understand the role of iron/iron oxide in this mechanism, RF and MF have been prepared because both derivatives exhibit significantly different magnetic properties. Using different techniques, including ultraviolet and visible (UV/VIS) spectrophotometry, atomic force microscopy (AFM), SQUID magnetometry, zeta potential measurements, and fluorescence spectroscopy, the physicochemical properties of derivatives as well as their impact on lysozyme amyloid fibrils have been examined. Differences in effective charges of RF and MF were correlated with different ferrous ions release. This allows us to propose the possible biophysical and biochemical mechanism of fibrillar structure destabilization based on the interactions with ferritin derivatives.

## 2. Results

UV-VIS spectroscopy was used to examine the protein and the iron content of the ferritin derivatives. The content was determined from the linear concentration-dependent curves (see Appendix A for protein and Appendix A for iron). The average ratio of iron atoms to protein moiety in the aqueous sample solution, according to Equation (1), is known as the iron loading factor (LF). The observed LF values are shown in Table 1.

The hydrodynamic diameter and zeta potential of the ferritin derivatives were examined using dynamic light scattering (DLS) and laser Doppler electrophoresis, respectively, and the results are summarized in Table 1.

Previously, the size of native apoferritin molecules was found to be ~10–12 nm (based on RTG diffraction) [20]. This slight difference between previously published size parameters may be caused by experimental conditions (concentration of particles, pH, and ionic strength) and different techniques used for measurement. The values obtained from the zeta potential correspond to the accepted negative total charge of proteins [21].

The dependence of magnetization on the applied magnetic field intensity measured at 290 K is shown in Figure 1. The decreasing magnetization trend points to the presence of a diamagnetic phase, which deforms the MF LF 160 curve. The observed behavior can be related to the presence of water or organic components (proteins) in the sample. As shown in Figure 1, the magnetic behavior of MF and RF is significantly different. The observed difference is due to the various synthesis procedures, which lead to the formation of different magnetic phases inside the MF/RF cavities. The formation of magnetite, maghemite, or hematite in MF and a ferrihydrite-like complex in RF is considered.

The colloidal stability parameter, i.e., zeta potential, provides information on the repulsion or attraction ability of the particles based on the surface charges on the particles dispersed in the solution and may be affected by interactions with other molecules. Zeta potential values at >±30 mV indicate good colloidal stability without a tendency to aggregation, which is desirable for biomedical applications. The positive or negative sign of the zeta potential value determines the sign of the total charge of the particles in the colloidal system. As documented in Figure 2, the zeta potentials of the mixtures of RF and MF with LAF of different mass ratios indicate that MF/RF remained stable.

The corresponding buffers, namely HEPES pH 7.4 (RF synthesis) and AMPSO pH 8.6 (MF synthesis), were used as solvents for the study of interactions with stable and insoluble LAF. The buffers ensured the stability of RF and MF and did not affect the stability (as well as the amount of LAF), as previously shown [19]. The zeta potential of stable insoluble LAF in HEPES and AMPSO pointed to a change in total surface charges. LAF dispersed in HEPES buffer exhibit a positive charge on their surface, whereas LAF dispersed in AMPSO are characterized by a negative charge (Figure 2). The presence of LAF in the mixture with MF did not change the total negative protein charge [21], although the colloidal stability decreased slightly with a decrease in the amount of MF in the mixtures. However, the colloidal stability of RF/LAF mixtures exhibits a significant decrease as a function of the RF concentration in the mixture. Despite the buffer effects on the electric double layers surrounding the objects, RF and MF caused a prevalence of a total negative charge in the mixture structures. The protein shell, apoferritin, with typical negative (alkaline) surface amino acid residues, might have contributed to this effect on pH.

AFM was used to visualize the morphology of LAF before and after incubation with ferritin derivatives (MF and RF). Figure 3A shows the typical fibrillar morphology of LAF. As can be seen, there are also fibrils with a length greater than 1 µm. The MF particles in Figure 3B are spheroidal in shape and form visible particle clusters. Figure 3C,D demonstrates that LAF incubation with MF/RF resulted in the almost complete destruction of fibrillar structures, and only small amorphous, spherical-like structures were observed.

For quantification of the anti-amyloid effect of ferritin derivatives on LAF, the thioflavin fluorescence assay (ThT assay) based on an increase in the fluorescence intensity of ThT upon binding to the structure of amyloid fibrils has been used [22,23]. Figure 4 illustrates the impact of RF and MF on the amount of LAF. The relative fluorescence intensity was calculated as a difference between the fluorescence intensity of ThT in the presence of pure LAF (taken as 100%), and the fluorescence intensity of ThT in the presence of LAF incubated with ferritin derivatives. It should be mentioned that the pure RF and MF solutions did not affect the ThT fluorescence. The results pictured in Figure 4 demonstrates that an increase in the weight ratios of RF/MF to LAF led to a decrease in ThT fluorescence, which can be interpreted as a decrease in the number of fibrillar structures in the solution. Unexpectedly increased fluorescence intensity at the ratio of 1:10 for both RF and MF may result from their higher density, which may induce the formation of large aggregates.

To examine the effect of LAF on the iron core of ferritin derivatives, the release of ferrous ions was monitored spectrophotometrically for 11 days. Figure 5 shows almost random wave-like dependencies for all samples. The Shapiro-Wilk test (α = 0.01) revealed the normality of the absorbance data for all analyzed samples, supported by Q-Q plots (Appendix A). A comparison of the average (Appendix A) and median (Figure 6) values indicate the extent of ferrous ions released during the interaction of ferritin derivatives with LAF. From this point of view, LAF behaves like a reducing agent of the ferritin mineral core, the same as vitamins B_2_ and C, presented in a recent study [7]. Induction of the release of ferrous ions from the ferritin shell during interaction with LAF can cause increased oxidative stress and more significant damage to the cell.

Except for the first day (time “0”), the total amount of ferrous ions released from RF is more significant compared to MF, regardless of the interaction with LAF (Appendix A). However, previous experimental findings regarding the interaction of vitamins B_2_ and C with MF and native ferritin (NF) showed a significant iron release in the case of MF compared to NF, for which RF should represent an analog. Due to the different LF/inorganic phase structures of model materials and specific properties of reagents (i.e., protective antioxidants and robust stable LAF), the ability to release iron ions must be related to other physico-chemical properties, especially redox potentials. Thus, the electron transfer will depend on all the characteristics of the reacting material in the specific medium.

An analysis of Spearman’s correlation coefficients revealed a strong correlation only for the RF and RF+LAF pairs. In all other cases, no strong correlations were found (Appendix A). Similarly, regression analysis did not reveal a linear dependence of iron release on time (Appendix A), indicating a more or less random process of iron release over time. However, as clearly shown in Figure 5 and Figure 6, the interaction of LAF with ferritin derivatives statistically significantly affects the higher rate of iron release.

## 3. Discussion

Ferritin is considered the precursor for iron phase transformation, and the transformation of its mineral core is associated with altered iron homeostasis and the development of neurodegeneration [24,25,26]. Recent studies confirmed the formation of various iron phases in ferritin, including a ferrihydrite-like complex in physiological ferritin and magnetite-based ferritin in the brain under pathological conditions [3,24]. However, it is not clear whether iron accumulation and ferritin transformation are the cause or consequence of pathological processes [27]. The dual effect of iron, essential protective behavior, and toxic, destructive behavior suggests similar behavior in complexes with (apo)ferritin. Previously, we showed the ability of vitamins B_2_ and C to reduce the mineral core of NF and MF as model systems of pathological ferritin [7]. In turn, the interaction of the amyloid *β*-peptide (A*β*)/fibrils with ferritin may induce the release of Fe^2+^ ions from ferritin molecules [28]. Everett and coauthors confirmed a similar conversion of the inert ferritin core to more reactive low-oxidation states upon the interaction of A*β* and ferritin [29]. In fact, the affected homeostasis of ferritin, the increased level of oxidative stress, and its involvement in aging and neurodegenerative diseases can be closely related. Modern trends in ferritin studies aim to achieve a precise regulation of iron duality by discovering molecular mechanisms to prevent disorders induced by fluctuations in extreme iron levels. Knowing the detailed mechanisms should allow one to apply or develop a suitable therapeutic approach. Therefore, we monitored the interaction effect between synthetic ferritin-based materials, i.e., RF (physiological-like mineral core)/MF (pathological-like mineral core) and lysozyme amyloid structures. The hydrodynamic diameter of both prepared materials was similar without observing a larger size, typical for aggregates, or disruption of the protein envelope secured by the low LF setting during synthesis (Table 1). Good colloidal stability (Table 1) and a distinguishable magnetic signal (Figure 1) allowed the use of RF and MF for testing. Specifically, different reaction conditions, according to procedures described elsewhere [21], led to an expected lower magnetization for RF (red line, Figure 1) and higher for MF (black line, Figure 1). Despite the oppositely charged LAF in HEPES (+) and AMPSO (−) related to the change in the electric double layers, the total surface charge of all RF/MF + LAF mixtures was negative. This effect may be related to the buffer property, which contributes to the redistribution of ions around stable insoluble LAF. MF was able to maintain a negative charge for almost all mass ratios. RF markedly decreased the stability of the RF + LAF mixture (Figure 2). The AFM images show a destructive effect of both ferritin derivatives on LAF after 24 h without the potentially reversible formation of fibrils induced by ferritin derivatives (Figure 3). The typical loss of fibrillar LAF morphology (Figure 3A) after exposure to ferritin derivatives (Figure 3C,D) supports our previous findings [19,30]. The quantification of the number of fibrils showed a decrease in fluorescence intensity caused by the elimination of LAF (Figure 4). The reduction in the number of LAF was more pronounced by increasing the mass concentration of RF/MF, as also observed in the latest study [19].

Following our previous studies [19,30], the missing explanation of the destructive effect led to an expanded investigation focused on the ferritin derivatives’ reduction ability of the iron core. First, aqueous samples of RF mimics of the physiological ferritin and MF, a model system of pathological ferritin with a low iron loading factor (LF), were prepared. Approximately five times lower LF than in the previous study should ensure the preservation of a stable protein structure [19]. Instead of common reducing agents (vitamins [7]), we monitored iron reduction after a large biopolymer (LAF) interaction. The analysis of the spectrophotometric results (Figure 5 and Figure 6) showed an increase in the reduction of the mineral core after interaction with LAF. A similar reductive effect of native ferritin cores was observed elsewhere after interaction with other molecules (e.g., vitamins, A*β* peptide) [7,28,29,31]. The known half-life of serum ferritin is short, approximately 30 h [32], and the half-life of plasma ferritin is about 12 h [33], due to the exposure of ferritin to degradation and recovery mechanisms in vivo. An extended monitored incubation time of 11 days was chosen to examine iron release kinetics in vitro. The results in Figure 5 suggest that ferrous ion scavenging is a spontaneous and energetically favorable process at room temperature. Wave trends probably matched iron reoxidation and reuptake by ferritin derivatives, i.e., iron flow. The ability and property of the ferritin channels support our interpretations. Hydrophilic 3-fold channels serve for iron ions input/output, and hydrophobic 4-fold channels for electron transfer. Following our works [19,30], the current results support the idea that the main factor contributing to fibril elimination is the redox potential of the iron core, i.e., electron transfer during the reduction of ferritin derivatives from Fe^3+^ to Fe^2+^. The reductive ability of lysozyme amyloid fibrils at pH~7.4 (HEPES) and 8.6 (AMSPO) is visible during iron flow simultaneously. The quantification of destruction is difficult due to the complexity of the fibril polymorphism (Figure 3A). A comparison of ferritin-like materials with different iron loadings, namely, ~500–700 [19,30] versus ~140–170 [30], suggests that the presence of iron is qualitatively the most important factor in fibril destruction. This is supported by our previous findings, where we showed the reduction and inhibition of LAF aggregation upon the interaction with pure magnetite nanoparticles composed of Fe^2+^ and Fe^3+^ ions [34] and iron-based magnetic fluids [35]. In this study, we showed that reconstructed ferritin (assumed to be Fe^3+^ core) was able to destroy fibrils and release Fe^2+^ easier than magnetoferritin (assumed to be Fe^2+^/Fe^3+^ core). This certainly deserves further research to analyze the effect of different iron compounds (source of Fe^2+^ and/or Fe^3+^ ions) on LAF destruction.

Various studies examining non-magnetic nanomaterials such as zeolite have confirmed the successful inhibition of fibrillation [36]. It follows that the nanodimensions of the materials are an equally significant factor. Due to their excellent biocompatibility, the ferritin derivatives used in the studies are predetermined for biomedical applications, including molecular and magnetic resonance imaging, drug delivery, and bioassays [37]. Moreover, the design of ferritin vaccines (spike-ferritin nanoparticle SpFN) has begun to enhance the SARS-CoV-2-specific durable adaptive immune T cell response [38]. In this study, we used ferritin derivatives to mimic physiological (RF) and pathological (MF) mineral cores of ferritin. Ferritin derivatives enable a detailed analysis of physico-chemical properties under precisely defined conditions that are not possible in(ex) vivo. Therefore, ferritin derivatives are irreplaceable in further research of iron-related physiological and pathological processes and iron-based biomedical applications.

## 4. Materials and Methods

### 4.1. Chemicals

Sodium hydroxide (NaOH), hydrogen peroxide ($H_{2}O_{2}$), ethanol ($C_{2}H_{6}O$), trimethylamine N-oxide (${\mathrm{Me}}_{3}\mathrm{NO}$), N-(1,1-dimethyl-2-hydroxyethyl)-3-amino-2-hydroxypropane sulfonic acid (AMPSO), equine spleen apoferritin (A3641/SLBD5084V), ammonium ferrous sulfate hexahydrate (${\left({\mathrm{NH}}_{4}\right)}_{2}\mathrm{Fe}{\left({\mathrm{SO}}_{4}\right)}_{2}.6H_{2}O$), N-(2-hydroxyethyl)piperazine- N-2-ethanesulfonic acid (HEPES), lysozyme isolated from hen egg white (E.C. number: 3.2.1.17, lyophilized powder, L 6876, ~50,000 units mg^−1^ protein), glycine ($NH_{2}CH_{2}\mathrm{COOH}$), sodium chloride (NaCl), Ferrozine Iron reagent, and thioflavin T (2-[4-(dimethylamino) phenyl]-3,6-dimethyl-1,3-benzothiazol-3-ium chloride) were obtained from SIGMA-Aldrich (Saint-Louis, MO, USA), Coomassie brilliant blue G-250 from Fluka (Buchs, Switzerland), hydrochloride acid (HCl) from ITES (Vranov n. T., Slovakia), potassium thiocyanate (KSCN) from Slavus (Bratislava, Slovakia) and phosphoric acid ($H_{3}{\mathrm{PO}}_{4}$) from Centralchem (Banská Bystrica, Slovakia). Demineralized (deionized) water was used to prepare aqueous chemical solutions throughout the experiment.

### 4.2. Preparation of Magnetoferritin (MF) and Reconstructed Ferritin (RF)

Ferritin derivatives, magnetoferritin (MF), and reconstructed ferritin (RF) were prepared by the in vitro controlled procedure [5,7,19,30]. First, the apoferritin solution was added to the 0.05 M AMPSO buffer and adjusted by the 2 M NaOH solution to a final pH of 8.6. The reaction solution was heated to 65 °C using a heater with a magnetic stirrer IKA C-MAG HS 7. After reaching a constant temperature, controlled additions of ferrous ions (0.1 M solution of ${\left({\mathrm{NH}}_{4}\right)}_{2}\mathrm{Fe}{\left({\mathrm{SO}}_{4}\right)}_{2}.6H_{2}O$) and oxidant (0.07 M solution of ${\mathrm{Me}}_{3}\mathrm{NO}$) were added into the reaction mixture in a stoichiometric ratio of 3:2 ten times during 100 min using syringes with constant stirring. All solutions were protected against air oxygen, displaced by inert nitrogen gas saturation for approximately 1 h. The reaction bottles were hermetically sealed to ensure anaerobic conditions. As a result of different amounts of added ferrous ions and oxidants, it was possible to prepare MF with a specific loading factor (LF—the average number of iron atoms per one biomacromolecule of apoferritin). RF synthesis was carried out similarly to MF, except for using a physiological temperature (37 °C) and buffer that maintained physiological pH (HEPES, pH = 7.4) during synthesis. The freshness of the sample was preserved by storing it in a refrigerator (4 °C) after synthesis. The RF and MF samples were post-processed by the standard freeze-drying method to obtain powders.

### 4.3. Preparation of Lysozyme Amyloid Fibrils

Lysozyme from hen egg white (lyophilized powder, L 6876; Sigma Aldrich, Saint-Louis, MO, USA) was dissolved in 0.2 M glycine buffer, pH = 2.2, to a final concentration of 10 mg/mL. The solution was incubated in a thermomixer C-MAG HS 7(IKA-Werke GmbH & Co. KG, Staufen, Germany) at 65 °C for 2 h with constant stirring at 1200 rpm. The ThT fluorescence assay and AFM confirmed the formation of lysozyme amyloid fibrils (LAF).

### 4.4. UV-VIS Determination of Loading Factor (LF)

Quantitative determination of LF was performed using UV-VIS SPECORD 40 (Analytik Jena GmbH, Jena, Germany) at 25 °C with a precision of about 1%. The iron-containing sample (MF: c_(protein)_ = 4.12 g/L, c_(Fe)_ = 0.0765 g/L, LF = 160; and RF: c_(protein)_ = 5.22 g/L, c_(Fe)_ = 0.0871 g/L, LF = 144), dissolved in concentrated hydrochloric acid, was oxidized using 3% hydrogen peroxide to a soluble ferric state at 50 °C for 30 min. The absorbance of the red ferric-thiocyanate complex (1 M KSCN), detectable at the wavelength of 450 nm, corresponds to the total iron concentration following the reaction: $2Fe^{3+}Cl_{3}+6KSCN\to Fe^{3+}{\left[Fe^{3+}{\left(SCN\right)}_{6}^{1+}\right]}^{3-}\left(red\right)+6KCl$. The protein concentration-dependent absorbance of a blue-colored complex between protein residues and the Bradford reagent was determined at a wavelength of 595 nm using the standard Bradford method.

Using the equation below (Equation (1)), the *LF* of magnetoferritin was calculated from the determined ratio of the mass concentration of iron atoms $c_{m}^{Fe}$ and the mass concentration of native apoferritin (NA), $c_{m}^{NA}$, in a given sample volume using the known molecular weights of native apoferritin (*M_NA_*, i.e., 481,200 Da, a value obtained from SIGMA-Aldrich, St. Louis, MO, USA) and the iron atom (*M_Fe_*, i.e., 55.845 Da).
(1)$$LF=\frac{C_{m}^{Fe}.M_{NA}}{c_{m}^{NA}.M_{Fe}}$$

### 4.5. Colloidal Stability and Total Charge Determination

The laser Doppler electrophoresis technology performed by ZS 3600 (Malvern Instruments, UK) measures the zeta potential (*ζ*) at 25 °C. Zeta potential, the parameter of colloidal stability, arises between the shear (slipping) plane of a solid particle in a conducting liquid and the bulk of the liquid. The parameter depends on the electrolyte concentration and pH. The application of an electric field causes the movement of particles in a liquid medium defined by a velocity or electrophoretic mobility described by Henry’s Equation:(2)$$U_{E}=\frac{2.\varepsilon .\zeta .f\left(K_{a}\right)}{3.\eta }$$

In Equation (2), $U_{E}$ stands for electrophoretic mobility, which depends on the strength of the electric field or voltage gradient, ε represents a dielectric constant of the medium, *ζ* is the zeta potential, $f\left(K_{a}\right)$ is Henry’s function, and *η* is the medium viscosity.

### 4.6. Measurement of Hydrodynamic Diameter

To measure the hydrodynamic diameter of the magnetoferritin colloidal solution, we used a ZetasizerNanoZS (Malvern Instruments, Malvern, UK). It uses dynamic light scattering (DLS), also known as photon correlation spectroscopy or quasi-elastic light scattering. This method allowed analysis of the scattered light intensity fluctuations from the MF (and RF) particles in solution (the buffer solution for MF was AMPSO, pH = 8.6, c = 0.05 M, and for RF it was HEPES, pH = 7.4, c = 0.02 M). These particles perform a Brownian motion. The measured rate of their diffusion in a liquid medium depends on the size according to the Stokes-Einstein Equation:(3)$$D_{T}=\frac{k.T}{6.\pi .\eta .R_{h}}$$

In Equation (3), $D_{T}$ represents the diffusion coefficient, *k* is the Boltzmann constant, *T* is the temperature, *η* is the solvent viscosity, and $R_{h}$ is the Stokes or hydrodynamic radius of the spherical particle measured in nm. Disposable polystyrene cuvettes were used in the protein data analysis mode at a temperature of 25 °C to measure the average hydrodynamic diameter <$D_{HYDR}$>. The size distribution of the relative particles number (%) was displayed using Zetasizer software with the hydrodynamic diameter representing the curve’s maximum. 

### 4.7. Analysis of Magnetic Properties by SQUID Magnetometry

The magnetic properties of the samples in the liquid state were analyzed using a SQUID magnetometer (Quantum Design MPMS 5XL, Quantum Design, San Diego, CA, USA). The hysteresis loops of the prepared samples were measured at a temperature of 290 K in the range of magnetic field induction up to 5000 kA·m^−1^.

### 4.8. Preparation of Mixed LAF-Magnetoferritin (MF) and LAF-Reconstructed Ferritin (RF) Samples

The stock suspensions of lysozyme fibrils were diluted in the appropriate buffers to achieve a fixed concentration of 2 mg/mL. MF and RF powders were added to fibril solutions at concentrations of 1, 2, 4, 10, and 20 mg/mL to reach the final amyloid: ferritin mass ratios of 1:0.5, 1:1, 1:2, 1:5, and 1:10, respectively. Colloidal suspensions were incubated for 24 h at 37 °C.

### 4.9. Visualization of Morphology and Size using Atomic Force Microscopy (AFM)

An atomic force microscope (Agilent Technologies, San José, CA, USA) equipped with PicoView 1.14.3 control software was used to reveal the morphology and size of the studied objects. A small amount of the sample suspension was dropped onto a piece of mica (PELCO Mica Sheets Grade V5, 15 × 15 mm^2^), allowed to adsorb on the mica substrate for 5–10 min, and then rinsed with ultrapure water and allowed to dry under ambient conditions. Topographic images were obtained in tapping mode using standard silicon cantilevers (Olympus, model OMCL-AC 160TS, Olympus Corp., Tokyo, Japan) with a resonant frequency of 300 kHz and a spring constant of 26 N/m. All measurements were carried out in the air, at ambient temperature and relative humidity between 30 and 40%. The freely available software Gwyddion helped with image editing (http://gwyddion.net/, accessed on 2 February 2020).

### 4.10. Thioflavin T Fluorescence Assay

Using the fluorescent dye thioflavin T, we used a fluorescence assay to determine the amount of LAF before and after the interaction with MF and RF. Thioflavin T (ThT) is a cationic benzothiazole dye that shows enhanced fluorescence upon binding to protein amyloid fibrils. The ThT dye was added to the LAF and LAF and RF/MF mixtures. The final concentrations of ThT and lysozyme aggregates were 20 and 10 µM, respectively. Fluorescence intensity measurements were performed on a 96-well plate using a Synergy MX (BioTek, San José, CA, USA) spectrofluorimeter. The setting of the excitation wavelength was 440 nm, and the emission was 485 nm. The necessary settings were adjusted as follows: excitation-emission slits to 9.0/9.0 nm and vertical offset of the upper probe to 6 mm. Triplicate fluorescence intensity measurements allowed the calculation of the final mean value for one mixture.

### 4.11. Time Dependence of Iron Release from Ferritin Derivatives

The ability of the ferrozine dye to bind Fe^2+^ ions allowed time-dependent spectrophotometric monitoring of iron flux from the shell of ferritin derivatives (RF and MF) with and without LAF. Sample absorbances were measured in triplicate after each 24 h for eleven days at 25 °C. The solutions contained 0.5 mL of sample and 1.47 mL of HEPES buffer. The prepared ferrozine solution with a volume of 0.03 mL was pipetted into Eppendorf each day just before absorbance measurement. The resulting absorbances of typical purple-colored ferrozine-Fe^2+^ complexes, detectable at a wavelength of 562 nm, were averaged. The regression equation of the constructed calibration curve allowed us to obtain the mass concentrations of released iron ions. Data processing of the time dependence of iron flow was analyzed using the Matlab 2021b software tool (Mathworks Inc., Natick, MA, USA).

## 5. Conclusions

Transformation of the inorganic phase in ferritins and the disruption of iron homeostasis are some of the still-unexplored mechanisms associated with various pathological processes, including neurodegenerative disorders. An iron storage metalloprotein, ferritin, expressed in almost all specific tissues in the forms of the spleen, heart, brain, or plasma isomer, serves as a model system in physico-chemical research or medical trends. The key goal of this experimental work was to assess the mechanism of mutual interaction between synthetic ferritin derivatives and well-studied LAF. The destruction effect of ferritin derivatives on LAF was confirmed. In particular, the size and amount of fibrillar structures were reduced, despite the low iron content in the ferritin derivatives. The results suggest that the main factor contributing to the elimination of fibrils is not only the presence of iron, but also the nanoscale dimensions of the particles, which increases the specific reaction contact area.

During reduction, LAF subsequently caused the flow of ferrous ions from the mineral core of both ferritin derivatives. The results indicated the necessity of the presence of free iron ions for fibril elimination. At the same time, released iron ions may be the reason for increased oxidation stress in neurodegenerative diseases. The cell response must be one of the DNA-regulated cellular safety mechanisms against nonspecific iron accumulation, e.g., overproduction of (apo)ferritin cages. Our results support the proposal to monitor brain iron and ferritin levels and determine the iron phase using physical methods for the early diagnosis of neurodegenerative diseases.

## Figures and Tables

**Figure 1 ijms-23-13926-f001:**
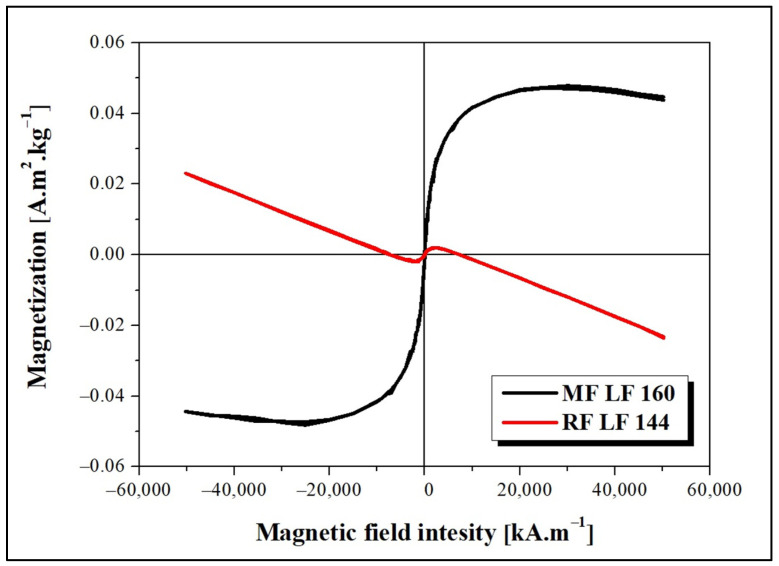
Dependence of magnetization on the magnetic field intensity of magnetoferritin (MF LF 160) and reconstructed ferritin (RF LF 144) at 290 K.

**Figure 2 ijms-23-13926-f002:**
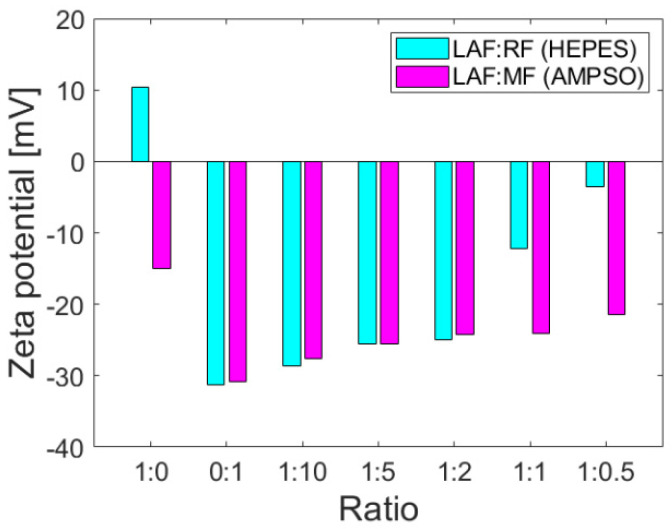
Zeta potential of the LAF and RF/MF mixture in various ratios. LAF concentration was fixed at 2 mg/mL.

**Figure 3 ijms-23-13926-f003:**
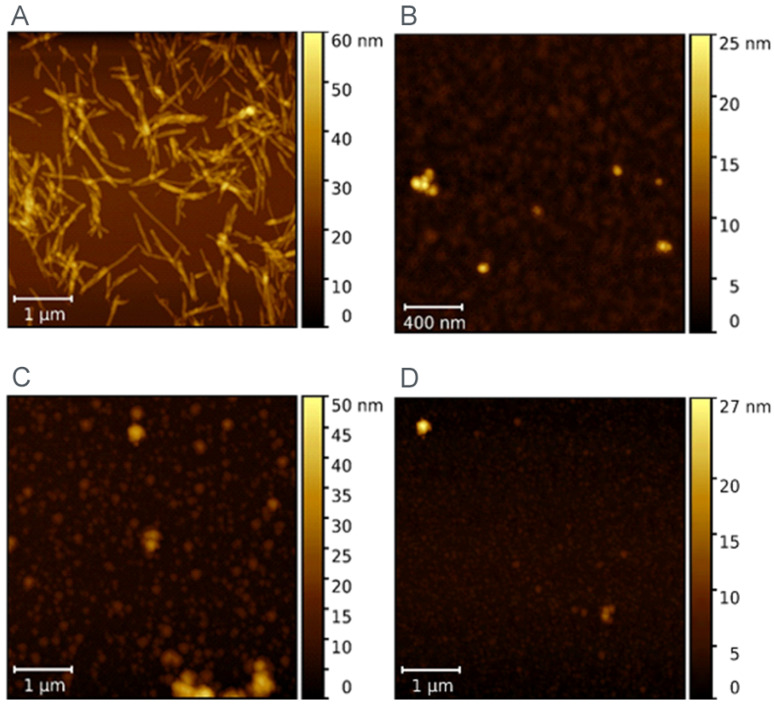
AFM image of the pure LAF (**A**) and pure MF (**B**), and visualization of the LAF after incubation with RF (**C**) and MF (**D**).

**Figure 4 ijms-23-13926-f004:**
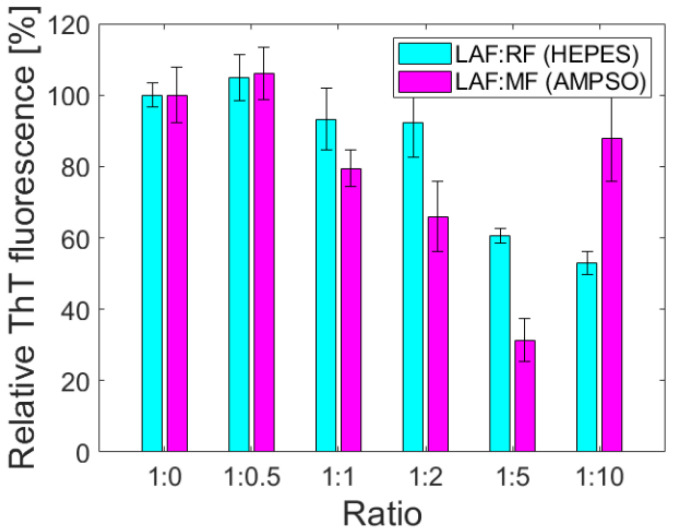
Concentration-dependent effect of ferritin derivatives on LAF monitored by ThT fluorescence assay. Examination of the destructive effect of ferritin derivatives on LAF: (cyan) destruction of LAF in the presence of RF with LF 144; (magenta) destruction of LAF in the presence of MF with LF 160. Concentration of LAF was fixed, 2 mg/mL.

**Figure 5 ijms-23-13926-f005:**
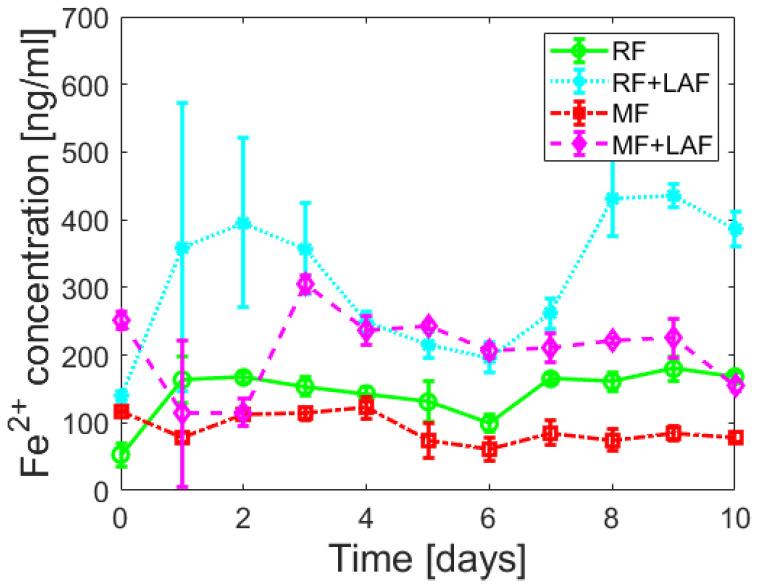
Time dependence of ferrous ion release (mean ± SD) from ferritin derivatives and during the interaction with LAF.

**Figure 6 ijms-23-13926-f006:**
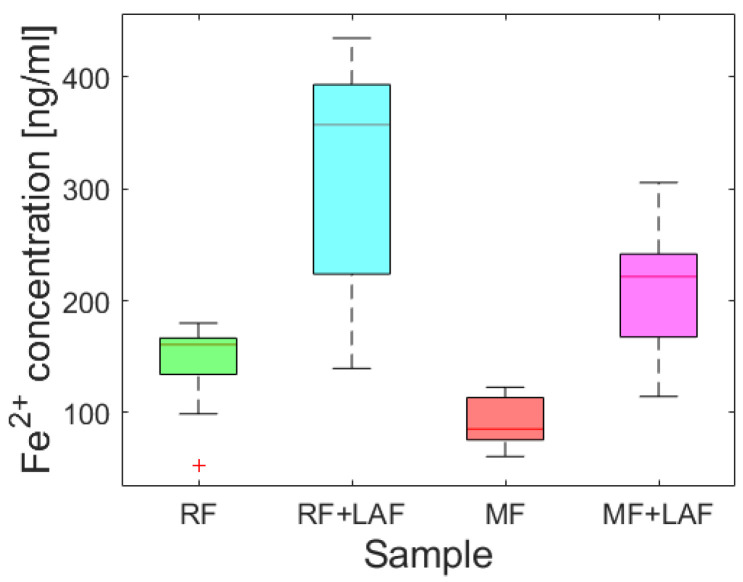
Median of ferrous ions released from the ferritin derivatives themselves and during interaction with LAF in 11 days.

**Table 1 ijms-23-13926-t001:** Loading factor, hydrodynamic diameter, and zeta potential of ferritin derivatives.

Sample	LF	<D_HYDR_> [nm]	ζ [mV]
RF	144	14.25	−31.2
MF	160	15.14	−30.8

## Data Availability

The data presented in this study are available on request from the corresponding author.

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
