# Peer review of "Destruction of Lysozyme Amyloid Fibrils Induced by Magnetoferritin and Reconstructed Ferritin"

_ijms, 2022, doi:10.3390/ijms232213926_

Round 1
Reviewer 1 Report (Previous Reviewer 1)
The authors should show the integrity of ferritin after the lyophilization step. I suggest a non-denaturing PAGE.
The destructive effect of the ferritins on lysosome amyloid fibrils was attributed to iron. This should be confirmed by incubating the fibrils with iron.
Author Response
"Please see the attachment."

Reviewer 2 Report (Previous Reviewer 2)
The authors have made extensive revisions and appear to have modified their paper in accordance with reviewers suggestions.
Author Response
Dear Reviewer,
Many thanks for your positive feedback.
Kind regards
Oliver Strbak
Round 2
Reviewer 1 Report (Previous Reviewer 1)
The answers are partially satisfactory
This manuscript is a resubmission of an earlier submission. The following is a list of the peer review reports and author responses from that submission.
Round 1
Reviewer 1 Report
The work studies the interaction between lysozyme amyloid fibrils with ferritins containing a limited amount of ferrihydrite or magnetite. The two ferritin types were studied for hydrodynamic diameter, zeta potential and magnetic properties, and the complexes with the fibrils were studied for zeta potential, iron release, atomic force microscopy and thioflavin fluorescence assay. The results indicate that both ferritin types slowly release iron which modifies the fibrils. Similar results were presented in a recent publication of the same group (ref 16). The work is not well presented and without novelties.
- the origin of ferritin is not stated. Probably it was a commercial horse spleen. The reasons for the low iron loading (144-160 Fe atoms per mol) are not defined and are probably far from the physiological ones.
- The characterization of the ferritins in table 1 and fig 2 should come first, and the rationale for the study of zeta potential is unclear, and it seems obvious that it changes with the buffer pH.
- figs 3-6 are redundant and the wave-like pattern of RF+LAF is not commented on. The kinetics of iron release last up to 11 days and seem non-physiological since the half-life of ferritin is much shorter than that, 1 or 2 days. Was all the iron released from the ferritins?
- Fig 7 is very similar to that of ref 16.
- A control with the addition of Fe(II) to the fibrils would be important.
Reviewer 2 Report
The paper does not go into any detail as to mechanism, either of the mode of reduction of ferritin iron on interaction with lysozyme amyloid fibrils or of the dissolution of the amyloid fibrils on exposure and complexation with ferritin.
It might appear that hydrogen peroxide formation either within ferritin or on the fibrillar surface could cause both fibrillar degeneration due to the close proximaty of the ferritin source and secondary formation of superoxide releading to ferritin iron mobilization, with catalysis of fibril breakdown, due Fenton reactions.
A simple way of testing would be to monitor the effects of catalase and or SOD to observe the effects of reducing solution H2O2 concentrrations.
Round 2
Reviewer 1 Report
The answers to the points I raised are not satisfactory and the modifications to the manuscript are only minor and they did not improve it significantly.
Author Response
"Please see the attachment."

Reviewer 2 Report
I have several concerns with the way some of the data have been presented in this paper and in a previous paper by the same group
In this paper fig 3 the solution concentration of Fe(II) is monitored at the times stated on the abscissa axis during an eleven day exposure of LAF to RF or MF. The concentrations of MF and RF are not clearly stated, but, given that the materials with the exception of ferrozine are all present for the entire 11 day duration of iron release, it is evident that after day 1 no additional iron is released in the case of RF and RF + LAF and with MF or MF + LAF no significant changes occur after day 0. Thus, the early release is not continued. Thus, it probably isn’t legitimate to aggregate the grouped results as in figure 4, as the key results are obtained in the first day or so.
Furthermore, as the solution concentration remains around an Fe(II) 2-3 uM for the duration of the experiment this could explain the puzzling absence of a time or concentration dependence of the ascorbate effects reported previously, Strbak et al IJMS 2020 21 6332. With this iron concentration in the the solution all ascorbate will have been oxidized within a few minutes, See Buettner 1986 FR Res Coms 1 349-353 ascorbate autooxidation in the presence of iron and copper chelates. So it is almost certain that is this is the explanation for the absence of any effects of ascorbate beyond the initial increases observed. If additional fresh aliquots were to be added on days 3, 5 and 7 it would be interesting to see if these cause additional iron release over a short time period.
I also have concerns about the results in figure 5. Comparing figure 5C and 5D shows a marked difference in the number of particles visualized in the fields. There is no indication whether these differences are statistically significant or simply the result of a random difference in the fields chosen by the microscopist. However additionally there is no proper indication of the time course of these changes. So it is not possible to say whether the changes produced by RF or MF on LAFs differ, the authors seem not to have wished to make any comment on this. However, I think this would be useful if properly quantified.
I am unclear with regard to figure 6 what it means, the changes in ThT fluorescence do not seem to be time related, wouldn’t a time course of changes be more appropriate rather than steady state, records?
However, the authors are right to emphasis the increased oxidation potential of pathological ferritins bound to fibrous materials in brain. Magnetite contains unchelated Fe (II) and which is which is capable of charge transfer between Fe(III) so that contact with this material can lead to the rapid dissemination of activated iron complexes disseminated throughout the ferrite crystal structure, unlike with the much more stable regular ferritin. Similar findings have been observed with magnetite minerals see Xiao et al BBA 2018 1862 p 1760-1769 and Huang et al Chemical reviews 2021 1212 8161-8233, it appears that sorbed Fe(II) complex with ligands
Nevertheless the presence of aberrant pathological forms of ferritin as discussed in this paper are important and should be investigated further, particularly
So I would like to see better quantification of the differential responses of LAF breakdown to RF and MF and also to regular Horse spleen ferritin LAF aggregates so that a convincing demonstration of a difference in oxidative response to MF v RF can be obtained.
Author Response
"Please see the attachment."

Round 3
Reviewer 1 Report
The work does not contain major novel data with respect to the ones already published by the same group, and the authors did not perform the experiments suggested by the reviewers. On top of this, I realised that the authors used lyophilised ferritins in their studies, although it is well-known that ferritin should never be frozen since this degrades the molecule with the formation of aggregates and precipitates. Thus, the experiments should be remade with ferritins that have never been frozen.
Reviewer 2 Report
The authors agree with the recommendation I made in my previous review, and say they are working on them, but currently have done nothing other than add the citations I have suggested. My opinion is that we should wait for the improvements to be available before publications, but this is up to the senior editors and their policies of producing interim and possibly inaccurate data.